# Sustainability of newborn screening for sickle cell disease in resource-poor countries: A systematic review

Chinwe O. Okeke[1,2]*, Chinedu Okeke[1,3], Samuel Asala[1,4], Akinyemi O. D. Ofakunrin[2], Silas Ufelle[5], Obiageli E. Nnodu[1,3]

1 Center of Excellence for Sickle Cell Disease Research and Training University of Abuja (CASRTA), Abuja, Nigeria, 2 Department of Pediatrics, Jos University Teaching Hospital, Jos, Nigeria, 3 Department of Haematology and Blood Transfusion, College of Health Sciences, University of Abuja, Abuja, Nigeria, 4 Department of Anatomical Sciences, College of Health Sciences, University of Abuja, Abuja, Nigeria, 5 Faculty of Health Sciences and Technology, Department of Medical Laboratory Science, University of Nigeria Nsukka, Nsukka, Nigeria

* chinwaem@gmail.com

**Data Availability Statement:** All raw data required to replicate this study are within the manuscript and its Supporting Information files.

## Abstract

Sickle cell disease (SCD) is a worldwide genetic blood disorder. Roughly 400,000 babies are born with SCD each year worldwide. More than 75% of these births occur in sub-Saharan Africa. The establishment of sustainable newborn screening NBS programs is an excellent approach to improving the health of persons living with SCD. The need to set up such programs in Africa cannot be overemphasized. However, initial implementation does not guarantee sustainability. More than 500 children with sickle cell anaemia (SCA) die every day due to lack of access to early diagnosis and related treatment. We systematically highlighted suggestions proffered so far, for the sustainability of NBS in low income, high burden countries. We searched online databases, PubMed, and Google Scholar for literature on sustainability of newborn screening (NBS) published between 2012 and 2022. Articles were included if they reported as outcome; sustainability, government participation, scaling up and expansion of NBS, improved patient enrolment in the newborn screening programe. Articles not suggesting same were excluded. Data were extracted from published reports. Primary outcome was government participation and enhanced patient enrolment in the NBS programe. Thematic content analysis was applied using inductive and deductive codes. We came up with 9 major themes. This study is registered with PROSPERO with registration number as CRD42023381821. Literature search yielded 918 articles (including manual searching). After screening, nine (9) publications were suitable for data extraction and analysis. Two more articles were added by manual searching, making a total of eleven (11) articles. The most frequently addressed core elements of sustainability in these papers were complete integration of services into national health care systems for sustainability of NBS programs in Low-income high-burden countries, funding and engagement from government partners from the very beginning of program development should be prioritized. Screening should be tailored to the local context; using DBS on HemoTypeSC could be a game changer for scaling up and expanding the newborn screening program in Sub-Saharan Africa.

**Funding:** The author(s) received no specific funding for this work.

**Competing interests:** The authors have declared that no competing interests exist.

## Introduction

Sickle cell disease (SCD) is a worldwide genetic blood disorder. A systematic analysis of the Global Burden of Disease Study states that 3.2 million people live with SCD, 43 million people have sickle cell trait and 176,000 people die of SCD-related complications annually [1]. The World Health Organization (WHO) as well as the United Nations (UN) designated SCD as a global health problem [2,3]. Roughly 400,000 babies are born with SCD each year worldwide, and more than 75% [3] of these births occur in sub-Saharan Africa [4]. Approximately 150,000 babies are given birth to annually in Nigeria. Up to 25% of Nigeria's population has the sickle cell gene.

Without effective and sustainable control strategies, this prevalence will increase exponentially [4,5]. SCD is caused by a point mutation in the beta-globin gene resulting in the substitution of valine for glutamic- acid to form sickle hemoglobin (HbS). HbS polymerizes under certain conditions of low oxygen tension, to create distorted, adherent, and less deformable red blood cells (RBCs). These RBCs have shortened lifespans, and are easily hemolyzed with a host of other pathophysiological effects that jointly contribute to the development of a group of acute and chronic clinical manifestations and complications and, often, early mortality. Foetal hemoglobin (HbF), is the predominant haemoglobin in new-borns, and the known most effective inhibitor of HbS polymerization [6]. As a result of this, infants with SCD are asymptomatic within the first 6 months of life until HbF levels decline to low levels. Early diagnosis before the preponderance of HbS is very important to make provision for early lifesaving interventions. Since SCD cannot be diagnosed by clinical signs at birth, newborn bloodspot screening (NBS) emerged many years ago to be a standard approach in many high-resource countries for identifying babies with SCD before the development of complications [7].

SCD mortality and morbidity have decreased during the first 20 years of life in high-income countries with well-established infrastructure for effective, universal newborn screening, early intervention, and comprehensive care, with less than 1% global disease burden and more than 90% of babies born with SCD surviving into adulthood [8]. In Africa, more than 500 children with sickle cell anemia (SCA) die every day as a result of a lack of access to early diagnosis and related treatment [9]. No country in Africa has implemented policies for universal screening [6]. The Republics of Benin and Ghana were the earliest two countries in Africa with comprehensive NBS programs [10]. Activities in other countries ranged from some NBS projects to pockets of pilot studies [11]. With keen awareness about the impact of SCD, there is optimism for increased progress in SCD newborn screening in the future. The establishment of sustainable NBS programs is an excellent approach to improving the health of persons living with SCD. The feasibility of setting up such programs in Africa cannot be denied, but initial implementation does not guarantee sustainability [6]. It is important to note that implementing NBS in low- and middle-income countries, is met with various challenges categorized as planning, leadership, education, medical, technical, and logistical support; policy development, administration, evaluation, and sustainability [12,13] and lack of funding by government. The design and execution of consistent operational processes of NBS from sample acquisition to laboratory testing and notification of results has been reported to be an intensive and challenging exercise. False perceptions about SCD, low turn-up for enrolment and follow-up, costs and regular maintenance of equipment, reliable access to reagents, and periodic unavailability of reagents leading to delays in testing are among the challenges encountered in the NBS programs in Africa. This systematic review aims to highlight some of the strategies that can be taken toward a sustainable NBS in resource-poor countries.

## Methods

We followed a step-by-step procedure for systematic review in the social sciences. The steps in this process were: (1) defining the research question; (2) defining the search terms; (3) selecting a database for the literature; (4) conducting the literature search; (5) developing inclusion criteria; (6) selecting literature using the inclusion criteria; (7) data extraction; and (8) aggregating and synthesizing the evidence. In this systematic review, we searched online databases, PubMed, and Google Scholar for literature on sustainability of newborn screening (NBS) published between 2012 and 2022. Articles were included if they reported sustainability, government participation, scaling up and expansion of NBS, improved patient enrolment in the newborn screening program, as outcome. Article that did not suggest same were excluded. Data were extracted from published reports. Primary outcome was government participation and enhanced patient enrolment in the NBS programme. We used thematic content analysis and a deductive and inductive framework to analyze data. PUBMED and GOOGLE SCHOLAR databases were searched in December 2022. Few manual searches were added. A total of 918 searches were gotten using the search terms "Sustainability of Newborn Screening programe for Sickle Cell Disease", "Sickle Cell Disease "AND "Newborn Screening", "POCT AND DBS AND "Newborn Screening". After deduplication, 572 searches were title screened. Articles were excluded, if the topic did not suggest; implementation, scaling up and expansion of newborn screening. A number of 442 articles were excluded after title screening and 130 articles were included in the study. These 130 articles were further subjected to abstract and full text screening. A total of 121 articles were further excluded based on the criteria listed in Fig 1. Only 9 studies provided a clear understanding of what is being done therefore contained useable information. There were very few literatures on sustainability of NBS for SCD in Africa. Andrew J.B. Fugard & Henry W.W. Potts [14] in their article: Supporting thinking on sample sizes for thematic analyses: a quantitative tool, recommended 10–100 sample size for secondary source in qualitative studies. We therefore added two more articles by manual searching bringing the total number of articles used in this review to eleven [11].

Critical Appraisal Skills Programme (CASP) [15] for qualitative studies was used to assess the quality of 10 articles. Critical appraisal for qualitative studies specifies ten criteria against which each study was screened. The number of questions answered with a "YES" represents the score out of 10. CASP for diagnostic studies was used to analyze the quality of 1 article. Critical appraisal for diagnostic studies has 12 questions but for this study 11 questions were applied because the 12th question requires a comment. The number of questions answered with a "YES" represents the score out of 11. The first column of Table 3 represents the 10 and 11 questions, the eleven selected studies had to be screened against.

From the CASP appraisal (Table 3), it can be seen that the average score for the 9 qualitative studies was 8/10 one article scored 6/10; the score for the only diagnostic study assessed was 10/11. This indicates an acceptable level of relevance and quality. For the diagnostic study the result of the test could not have been influenced by the results of the reference standard and so the response to this question was No and yet a positive response adding to the count. None of the qualitative studies reported on obtaining ethical clearance except for the diagnostic article. This could have been due to the fact that the studies did not require ethical clearance since most of the studies involved the use of questionnaires.

### Data extraction

Data extraction was carried out using extraction form adapted from the extraction form used by Hoogland et al, [16], a form was created for the extraction of data from the chosen publications. Using a form made it possible to collect comparable data from the chosen publications ([17].

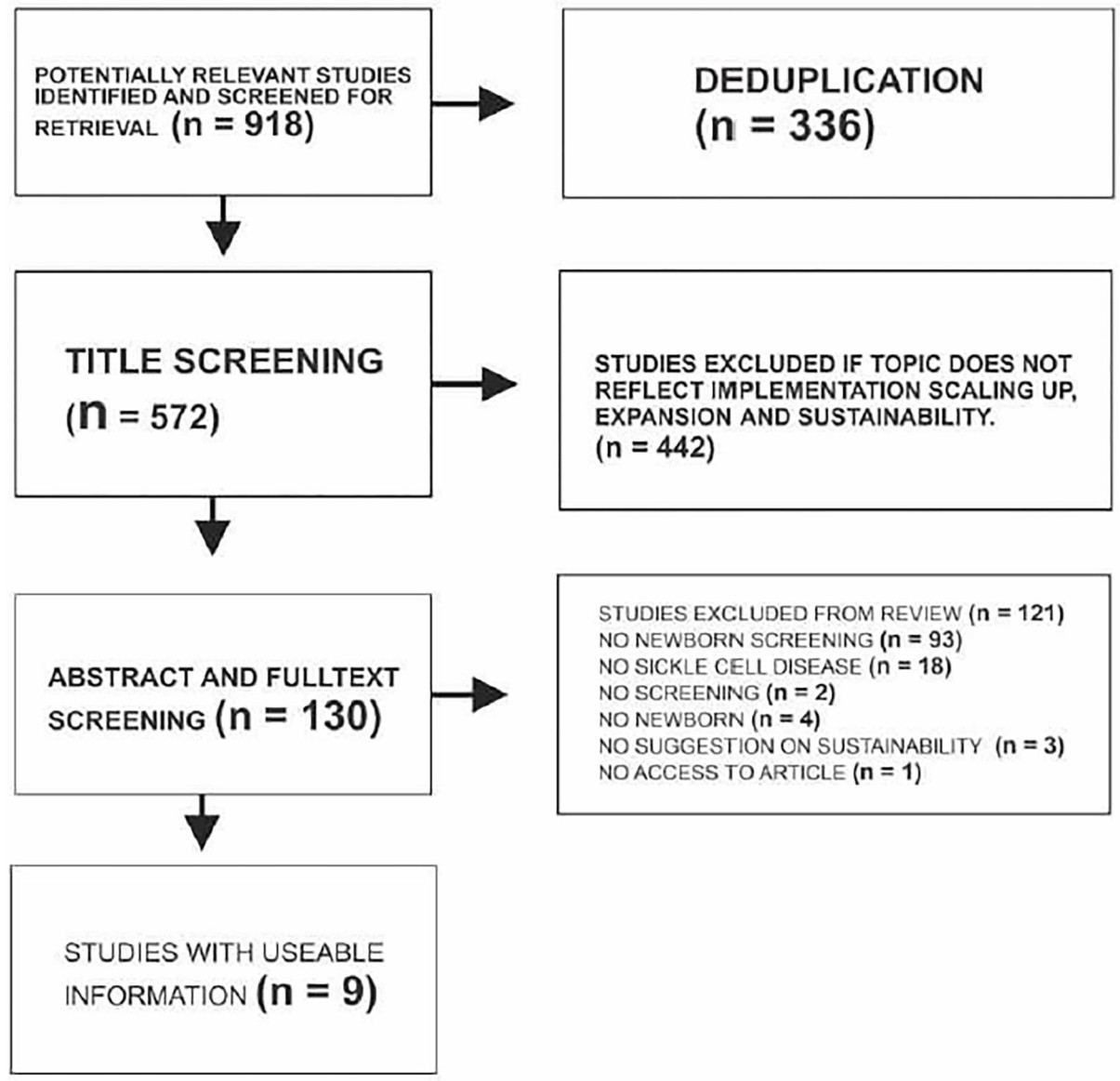

**Fig 1. PRISMA flow diagram.**

## Data analysis

Two reviewers conducted the searches and data extraction. Data extraction was done using data extraction form (Table 1). Thematic content analysis was applied. The extracts were coded, using inductive and deductive approach; the codes were further aggregated into themes. Nine (9) major themes were identified as shown in Table 2. Only three articles delved in-depth on the sustainability of Newborn screening programs in resource-poor countries. The remaining articles mentioned important aspects of sustaining NBS in resource-poor countries. The most frequently addressed specific (core) elements of sustainability in these papers was the complete integration of services into national health care systems. All the Eleven (11) articles, reported on the critical role of Government involvement in driving the NBS, hence the integration of services into national health care system, to increase service availability

**Table 1. Shows data extraction form.**

| S/N | References | Investigated newborn intervention | Research method | Problems/Challenges highlighted | Progress/Solutions proffered | Relevant endpoints |
|---|---|---|---|---|---|---|
| 1. | Enablers and barriers to newborn screening for sickle cell disease in Africa: results from a qualitative study involving programs in six countries [6]. | Development, implementation and sustenance of NBS Programs | Qualitative | Challenges with consistent access to reagents, equipment maintenance. Testing delayed due to frequent reagent shortages | National Health Authorities role. Government service delivery units. Government active and passive support. Staffing, Government employed workers, Teams offering specialized care to the patients. Availability and accessibility of clinical centers of excellence. Providing holistic preventive and treatment services for individuals with sickle cell disease. Fully integrate NBS workflows into routine health system. Data collection and management system that support workflows. Adequate record keeping either manual or digital or a hybrid of the two methods. Reliable access to reagents. NBS services provided free. External funding | Ensures sustainability. NBS programs for SCD can be feasibly and successfully implemented in Africa, as indicated by the large number of babies screened and the protracted screening period (for example, more than 25 years) that was exhibited in several programs. |
| 2 | Empowering newborn screening programs in African countries through establishment of an international collaborative effort [11] | Workshop to investigate new information and create beneficial partnerships for enhancing child health | Qualitative | All programs got some sort of outside support, which is defined as funds from organizations located abroad. The relationship between fertility rates and poverty which is very high in Africa is noted to compound the significant economic burdens associated with delivery of health care generally. A dearth of medical professionals with genetics training, familiarity with NBS and SCD, and ability to support families and counseling. The high cost of reagents was a major limitation to the success of NBS. | Screening and other case-specific data are critical for effective patient tracking and program evaluation. Development of longitudinal registry of patients with sickle cell disease. Data collection and use in policy development. There is a recommendation to include NBS to the ongoing Bacillus Calmette–Guérin (BCG) immunization program. Point-of-care NBS technology has been piloted, and recommended. Success of the NBS program in some countries attributed to efficient resource sharing with ongoing immunization programs or HIV early intervention initiatives. Cost effectiveness Assessment for quality improvement. | Across Africa NBS programs vary in size, history, and degree of success. NBS sustainability depends on local MoH approval and support since it necessitates integration of all system components within local (jurisdictional) geographic, economic, and political restrictions. |
| 3 | A Quality Improvement Collaborative to Improve Pediatric Primary Care Genetic Services [18] | 6-month QIC that included regular educational opportunities, access to genetic professionals, and performance feedback | Qualitative | Though many genetic diseases have particular health supervisions guidelines, it is unknown how frequently pediatricians follow these or the best way to guarantee guideline implementation. | Creation and discussion of family histories. Appropriate clinic registry Standard of care guide lines Teams offering specialize care Health authorities' roles | Increasing and maintaining adherence to process standards for the diagnosis and treatment of genetic disorders shows potential, according to a national QIC. |
| 4 | Newborn Sickle Cell and Thalassemia Screening Programme Automating and Enhancing the System to Evaluate the Screening Programme [19] | construct a fresh automated system to guarantee that screening program fulfills its responsibility to assess the program's efficiency | Mixed Method | Absence of proper oversight of the baby to ensure safe handover from screening into treatment services. frustration amongst data providers due to repetition of data entry, and poor response | Active system that provides oversight of screen positive babies identified by the Screening Programme as they are transferred from screening to clinical services. Prompt identification of newborns with sickle cell disease. Timely movement into clinical care. Beta thalassemia major, identified along the way. Generation of good quality datasets. | Tracks infants who test positive for SCT as they move from screening to clinical services. SCD positive newborn must be enrolled in pediatric care by 90 days of age and parents must get screen positive results by 28 days. |
| 5 | Implementing newborn screening for sickle cell disease as part of immunization programs in Nigeria: a feasibility study [20] | Hemoglobin Typing, using HemoTypeSC POCT | Qualitative | Due to screening technologies' high cost and technology investment, early detection and enrollment of sickle cell disease patients into comprehensive care programs have been severely hampered in countries with limited resources. Methods that are affordable, dependable, and precise are required. | Integrating NBS into existing primary health- care immunization programs can rapidly be implemented with limited resources, Using point of care testing to reduce cost Availability of Prenatal and postnatal education and counseling to parents and relatives. Cost-effectiveness of newborn screening, depends largely on the affordability, and accessibility of the tests, care, and follow-up early detection by newborn screening needs a helpful and efficient public health Set-up to be managed. Acknowledge limitations of traditional screening methods in resource limited settings. | Sickles CAN and HemoTypeSC test results, were 1001% consistent with HPLC test results. |

*(Continued)*

**Table 1.** (Continued)

| S/N | References | Investigated newborn intervention | Research method | Problems/Challenges highlighted | Progress/Solutions proffered | Relevant endpoints |
|---|---|---|---|---|---|---|
| 6. | Family, Community, and Health System Considerations for Reducing the Burden of Pediatric Sickle Cell Disease in Uganda Through Newborn Screening [21] | Focused discussion on SCD screening, including designing research on assessing ways to effectively notify parents about test results as well as enrolling affected newborn for comprehensive care | Qualitative | Need for timely parental notification of SCD aimed at early entry into special and continued care. Poor parental understanding about SCD. Need for expansion of SCD care and a system of regional hubs of SCD clinics and smaller local health facilities by general pediatric staff. Need for excellent diagnostic capabilities | Taking advantage of the existing Set-up for hub-based sample collection and delivery to a testing laboratory. Early parental notification of SCD intended at early enrollment into special and sustained care are critical for program success. The Uganda Ministry of Health (MoH) and other significant players are dedicated to the success of the newborn screening program and medical treatment for SCD in order to decrease child mortality linked with SCD. Deployment of community workers to parents' homes to inform them about sickle cell results and for other follows up plans. Village health workers could be trained to perform counseling and guidance to parents to encourage visits to health facilities. An efficient, effective and scalable process to link screening, parental notification, and timely entry into standardized care is needed. Novel mobile health (mHealth) technologies could help reach parents with screening results. Education. Sharing information on program creation, problem solving, and lessons learned is essential. To reduce the burden of disease, scholarly and advocacy enterprises must be synchronized internationally. Research must pay attention on recognizing the social, practical, and financial difficulties that prevent kids from receiving timely and ongoing care for SCD. | Both healthcare professionals and decision-makers concurred that one of the most economically advantageous ways to make the most of the scarce resources is to expand NBS services for SCD to cover other illnesses. The suggested neonatal conditions that might be mentioned were examinations for HIV, hearing issues, Down syndrome, and other physical anomalies including talipes. |
| 7 | Perspectives on Building Sustainable Newborn Screening Programs for Sickle Cell Disease: Experience from Tanzania [22] | Efforts and experiences of NBS for SCD in Tanzania to build a sustainable program in a resource constrained environment | Qualitative | Laboratory diagnostics, specialized training to health care providers and the need for high level advocacy to ensure government support in the implementation. | Begin the programs in well-resourced Regional Referral Hospitals before moving to district hospitals. Complete integration of the services in countries' health care systems to enable the coverage, accessibility and affordability of the service. Encourage doctors and nurses to go for training related to newborn screening. Educating and counseling parents on the need for screening' Availability, affordability and accessibility of comprehensive care services. Establish sickle cell clinics in primary health care facilities. Providing training for specialized care for SCD at all levels of health care. Do follow up for patients who do not come to the clinic. Expand NBS to include other diseases. Engage with private sector. Advocacy. | Demonstrated both the prospects and areas of interest in the implementation and sustainability of the NBS services in low resource settings. Full integration of the services in health care noted as a key area |
| 8 | Using dried blood spot on HemoTypeSC™, a new frontier for newborn screening for sickle cell disease in Nigeria [23] | Using DBS to run on HemoTypeSC Strip | Diagnostic | A need for a poor resource setting tailored screening method for better efficiency in mass screening settings | The traditional point-of-care HemoTypeSC test is as accurate when using dried blood spots. Mass newborn screening could be more realistic and affordable using DBS on HemoTypeSC for SCD. Proves to be cheaper and less laborious than current methods in use. Does not require much technical skill to be used. Has the potential to scale up and expand screening in the African region with limited resources. | DBS on HemoTypeSC offers hope for easy scaling up of newborn screening for SCD in resource poor settings |

(*Continued*)

**Table 1.** (Continued)

| S/N | References | Investigated newborn intervention | Research method | Problems/Challenges highlighted | Progress/Solutions proffered | Relevant endpoints |
|-----|-----------|-----------------------------------|-----------------|--------------------------------|------------------------------|--------------------|
| 9. | Pathways to Progress in Newborn Screening for Sickle Cell Disease in Sub-Saharan Africa. Lewis HSU et al 2018 [24] | The government of SSA countries have not recognized the complex relationship between sickle cell disease, infectious diseases and other NCDs. | Qualitative | Universal newborn screening for SCD implementation has been successful in countries with low prevalence of SCD compared to SSA countries having 85% of the total burden of SCD. There is an absence of cost effectiveness study of newborn screening for SCD in West Africa. | Availability of facilities for proper diagnosis and SCD management, Data Availability on frequency, clinical course, morbidity and mortality as well as record of birth and all deliveries, are resources to convince African governments and donor agencies to focus on a comprehensive care and management plan. WHO has plan to fund major prevention of SCD in Africa, but various governments have to show satisfactory interest in having practicable and comprehensive care plans on ground. A cheap method in generating satisfactory data on SCD mortality in children in order to convince policy makers in Africa to establish primary care is by testing blood also collected for some other studies like HIV, Malaria, and malnutrition. Monitoring and assessment are essential parts of any program on SCD. Development of National SCD NBS policy and protocol that is locally applicable and suitable considering available resources. Supported by the Ministry of health that ensure successful implementation and extension of NBS program. Public engagement and advocacy UN treaties and conversions on the rights of the child of which most countries are signatories, can be used to woo policy makers buy into SCD NBS. Public private partnerships (PPP) + Taking advantage of NGO Application of mobile phone technology. | Described the challenges and prospects for NBS for SCD, and why Evidence- based interventions has not been adopted in Africa. |
| 10. | Newborn Screening for Sickle Cell Disease in the USA and Canada. Nura El-Haj and Carolyn Hoppe 2018 [25] | The historical context, contemporary strategies, and techniques for screening infants for sickle cell disease (SCD) in the US and Canada were compiled in the current review. | Qualitative | Up until the year 1999, all except 9 states in the USA had established universal NBS for SCD. A national process to ensure uniformity among states in quality of testing, interpretation of results, collection of outcome data was lacking | SCD Screening; a response to mounting political pressure by African American Advocacy groups in the 1960's State recognized SCD as a public health issue of significance. By 1972 the National Sickle Cell Anaemia Control act was passed by the congress. DBS introduced as effective method for collection and testing of blood samples from newborns. NBS programme by 1975, funded over 250 general screening programmes, 41 sickle cell centers and clinics as well as 69 research grants and contracts, with numerous locally supported screening, education and counseling clinics. Following the success of evidence-based interventions; the National Institute of Health convened a conference that prompted unanimous support for universal NBS for SCD. By 2006 all 50 states and many U.S regions adopted Universal newborn screening for SCD with Hb SS, Hb SC, and Hb S/βthalassemia as core panel. Targeted NBS metamorphosed to Universal Newborn screening | Efforts should be geared towards standardizing nomenclature and collection of resultant data via the development of linked registries as initial step in achieving the long-term follow-up goal. |

(Continued)

**Table 1.** (Continued)

| S/N | References | Investigated newborn intervention | Research method | Problems/Challenges highlighted | Progress/Solutions proffered | Relevant endpoints |
|---|---|---|---|---|---|---|
| 11 | Implementing newborn screening for sickle cell disease in Korle Bu Teaching Hospital, Accra: Results and lessons learned [26] | Determination of the practicability and challenges involved in implementing NBS for SCD in Ghana's largest public hospital | Qualitative | Implementation, improvement and sustainability challenges confronting the newborn screening programme. | Persistent agitation, led to government intervention, hence expansion of the pool of professionals. Digital mobile application for quicker test result collection. Task- shifting to mitigate critical health worker shortage. Building a dedicated team of DBS sampling. Addressing bottlenecks in screening supplies. Widespread implementation of digital addresses would enhance; tracking of screen positive babies, quality of care, morbidity etc. Creating small teams whose primary responsibility is NBS at the primary health centers. Development of point of care technologies for sickle cell diagnosis. Easy access to healthcare services. Education for all concerned. Efficient Longitudinal data capture. | NBS interventions must be adapted to local conditions and incorporate support, scaling-up, and sustainability measures. In a resource-limited setting, leveraging existing capacity is very necessary. The National Sample Referral Transport Network program from early infant HIV detection (EID)19 was modified to provide infrastructure for hub-based sample collection and transport to a testing laboratory. CD screening began in regions with higher trait prevalence, as earlier documented by the MOH. Extension of SCD care by general pediatric personnel, together with a network of regional hubs of SCD clinics and smaller neighborhood health centers |

**Table 2. Themes with article numbers corresponding with extraction form (Table 1).**

| Extracted aspects of sustainability | Article Numbers in the extraction form |
|---|---|
| Complete integration of services into national health care system | 1,2,3,4,5,6, 7,8,9,10,11, |
| Suitable National SCD protocols | 2, 5, 6,9,10,8,2,7,1,11 |
| Expanding NBS for SCD to include other diseases | 2,5,6,7,8,9 |
| Efficient public health interventions | 9, 4, 5, 6, 11,7,10 |
| Quality improvement collaboration | 3,2, 9,10 |
| Data capture and management | 1,2,3,6,7,9,4,10 |
| Targeted NBS | 1,2,6,7,9,10 |
| Advocacy | 1, 5, 6,7,8,10,11 |
| Funding | 1,7,9,4 |

affordability and coverage. Ten (10) articles reported on the Development of National SCD NBS protocol that is locally applicable and suitable, considering available resources, six (6) on expanding NBS to include other diseases, Seven (7) on efficient public health interventions, Four (4) on quality improvement collaborations (QIC), Eight (8) on efficient longitudinal data capture, Six (6) on targeted NBS for SCD, Seven (7) on advocacy and four (4) on Funding. This study is registered with PROSPERO, with the registration number: CRD42023381821.

## Results

Nine (9) publications were initially found suitable for data extraction and analysis, after manual searches were made, two additional articles were added making a total of eleven (11) articles. The characteristics of the included studies are shown in Table 3.

The following findings were made.

### Complete integration of services into national health care system, to increase service availability affordability and coverage

The active support of the government in the NBS programe was the specific (core) aspect of sustainability that was mostly discussed in these articles. All the Eleven (11) articles mentioned directly or indirectly the pivotal role of countries' governments in the sustainability of NBS for SCD. The role of the government in staffing, accessibility, affordability of test and follow up was widely discussed. A widespread implementation of digital address system by the government was said to enhance tracking of screen positive babies [26]. A qualitative descriptive study using semi-structured interviews among program leaders in various African countries stated that prioritization of government partner funding and engagement from the very beginning of program development is crucial for sustainability [6]. According Archer et al [6], all NBS programs got some sort of outside support. Other sources of support included Foundations, non-governmental organizations, businesses in the private sector, and governments of other nations. The need to scale back or end the program in some circumstances was caused by the loss of external financing, external funding was therefore typically seen as a "double-edged sword," as it was important for some programs to materialize but also made it more difficult to achieve long-term sustainability because of the non-feasibility of permanent funding from outside sources. It is advocated that going forward from the very beginning of program development, government-partner funding, and engagement should be prioritized. For a program to be sustained over time, it may take more than just proving its viability and accumulating data to prove that it is linked to beneficial outcomes in terms of outputs and health. Full

**Table 3. CASP appraisal table [15].** The appraisal of the Eleven articles that were used in the review using CASP. No article was dropped as the Eleven articles met the criteria of methodology, adequacy, and relevance.

| Questions | Archer et al., 2022 [6] | Lewis HSU et al., 2018 [24] | El-Haj N, Hoppe CC2018 [25] | Rinke et al., 2016 [18] | Coppinger et al., 2019 [19] | Daniel et al., 2019 [27] | Nnodu et al.,2020 [20] | Green et al., 2016 [21] | Bukini et al.,2021 [22] | Segbefia et al., 2021., [26] | Questions | Okeke et al.,2022 [23] |
|---|---|---|---|---|---|---|---|---|---|---|---|---|
| Was there a clear statement of the aims of the research? | Y | y | y | Y | Y | Y | Y | Y | Y | Y | . Was there a clear question for the study to address - | Y |
| Is a qualitative methodology appropriate | Y | y | y | Y | Y | Y | Y | Y | Y | y | Was there a comparison with an appropriate reference standard? | Y |
| Was the research design appropriate to address the aims of the research? | Y | y | y | Y | ? | Y | Y | Y | y | Y | Did all patients get the diagnostic test and reference standard. | Y |
| Was the recruitment strategy appropriate to the aims of the research? | Y | y | y | Y | Y | Y | Y | y | Y | Y | Could the result of the test have been influenced by the results of the reference standard | N |
| Was the data collected in a way that addressed the research issue | Y | y | ? | y | Y | y | y | y | y | y | Is the disease status of the tested population clearly described | y |
| Has the relationship between researcher and participants been adequately considered | ? | ? | ? | ? | ? | ? | ? | ? | ? | ? | Were the methods for performing the tets described in sufficient detail Are the sensitivity and specificity and/ or likelihood ratios presented | Y Y |
| Have ethical issues been taken into consideration? | N | ? | N | N | N | Y | N | N | N | ? | Are there confidence limits | Y |
| Was the data analysis sufficiently rigorous? | Y | y | N | Y | Y | Y | Y | Y | Y | Y | Can the results be applied to your patients/ the population of interest? | ? |

*(Continued)*

**Table 3.** (Continued)

| Questions | Archer et al., 2022 [6] | Lewis HSU et al., 2018 [24] | El-Haj N, Hoppe CC2018 [25] | Rinke et al., 2016 [18] | Coppinger et al., 2019 [19] | Daniel et al., 2019 [27] | Nnodu et al.,2020 [20] | Green et al., 2016 [21] | Bukini et al.,2021 [22] | Segbefia et al., 2021., [26] | Questions | Okeke et al.,2022 [23] |
|---|---|---|---|---|---|---|---|---|---|---|---|---|
| Is there a clear statement of findings? | Y | y | y | Y | y | y | Y | y | y | Y | Can the test be applied to your patient or population of interest | Y |
| Did the researcher discuss the contribution the study makes to existing knowledge? | y | y | y | y | y | y | y | y | y | y | Were all outcomes important to the individual or population considered | Y |
| Score out of 10 | 8 | 8 | 6 | 8 | 7 | 8 | 8 | 8 | 8 | 8 | Score out of 11 | 10 |

integration of the services in countries' health care systems to facilitate the coverage, accessibility, and affordability of the service, establishment of centers of excellence, were suggested. NBS sustainability depends on local Ministry of Heath approval and support since it necessitates the integration of all system components within local geographic, economic, and political restrictions [11].

## Development of National SCD NBS protocol that is locally applicable and suitable, considering available resources

A total of ten articles alluded to development of SCD NBS protocols that are locally applicable and suitable, considering available resources. Targeted newborn screening for SCD was practiced in the United States of America for 20 years before the implementation of universal NBS for sickle cell disease in 50 states of the Nation [25]. Five publications, reported on the importance of leveraging on POCT technology applications to scale up NBS for SCD in Africa and other resource limited settings. Okeke et al., [23], discovered that dried blood spot can be used on a POCT (HemoTypeSC) and accurate result obtained. Hence, using DBS on POCT(HemoTypeSC) could be a game changer for mass newborn screening in places like Africa due to its cost effectiveness. This discovery goes a long way to satisfy the need for affordable, reliable, and accurate testing methods that can be integrated into existing primary health- care immunization programs. Methods that are affordable, dependable, and precise such as the use of point-of-care testing devices are required. Examples of such devices include Sickles CAN and HemoTypeSC and the test results obtained are 100% consistent with that of HPLC [20]. The conventional point-of-care HemoTypeSC test is as accurate when using dried blood spot.

## Expanding NBS to include other diseases

Six (6) articles advocated A hub-based collection of specimen and result delivery, for HIV, Malaria, malnutrition, other congenital diseases and SCD for cost effectiveness. Both health-care professionals and decision-makers agreed that one of the most economically advantageous ways to make the most of the scarce resources is to expand NBS services for SCD to cover other illnesses. The suggested neonatal services were HIV testing, screening for hearing, Down syndrome, and other physical anomalies like talipes, cleft palate. Expanding the

screening services, will make provision for conducting a variety of tests under one roof, making room for checks for other disorders like HIV using the same DBS samples [22]. In practical mass screening situations, such as immunization centers, there are still gaps in the implementation of the standard POCT as it were. The use of standard HemoTypeSC POCT will not be sufficient when a battery of tests must be performed as it is carried out in industrialized nations. The application of DBS on HemoTypeSC will bridge these gaps. Using DBS on HemoTypeSC is a poor resource setting tailored screening method for better efficiency in mass screening settings for SCD [23]. Due to screening technologies' high cost and technology investment, early detection and enrolment of sickle cell disease patients into comprehensive care programs have been severely hampered in countries with limited resources.

## Efficient public health interventions

Seven (7) articles reported on the need for efficient public health interventions as a motivation for patients' compliance, in the NBS programe which encourages sustainability.

Five (5) of the articles, stated the need for prompt notification of results as a prerequisite for early enrolment into comprehensive care, two articles reported on adequate process linking screening, parental notification and patient enrolment. Early diagnosis, newborn screening, and genetic counseling are essential components of effective SCD care, nutrition and nutrition education need to be integrated, policies on genetic counseling and screening should be legislated and given the needed executive resources to be implemented [24], Early parental notification of SCD intended at early enrollment into special and sustained care are critical for program success [21]

## Expansion of services to primary and secondary health care facilities

Families must travel a substantial distance to obtain care because the majority of specialized care for SCD is located in regional referral hospitals (RRHs). Poor clinic attendance for some families was correlated with a lack of funding for transportation. For the RRHs to provide specialized treatment for SCD patients who are not managed at primary and secondary health facilities, services for SCD will be introduced in primary and secondary health facilities. This will lighten the burden of care [22]. As a way of sustaining comprehensive care services for SCD, it was suggested that staff working in primary health centers be given the necessary training. This would facilitate the distribution of work across the various levels of care and may help to reduce the stress on regional referral hospitals and zonal hospitals [21,22]. The goal of all programs was to minimize the amount of time between sample collection and when the families were informed of the results. Finding families to share laboratory data was one frequent cause of delays in the NBS workflow; some families could not be reached by phone, necessitating in-person visits that took a lot of time and were not always effective. In one program, the authors reported that specimens had to be driven from the birthing sites to the laboratory across a distance covered in 7 hours because the laboratory was in a separate city from the birth centres. Another program flew samples to the NBS program laboratory in another nation in a sealed container at 4˚C. Most programs aimed to fully incorporate NBS workflows into standard health system operations [11]. Integrating NBS into existing primary health-care immunization programs can rapidly be implemented with limited resources, using point-of-care testing [20]. Using DBS on HemoTypeSC promises to accelerate this initiative [23].

## Task shifting

Village health workers could be trained to perform counseling and guidance to parents to encourage visits to health facilities.According to Segbefia et al., [26], the overall shortage of

nursing personnel across Africa is will predictably worsen by 2030. It is therefore advised to use a task-shifting technique for the effective administration of NBS for SCD, whereby NBS training was given to preventative health care assistants, freeing up the few highly competent nursing staff members to do more difficult patient-related responsibilities. Critical health worker shortages can be lessened in resource-constrained environments when task-shifting within health care teams is performed methodically, with the right training and staff motivation.

## Quality improvement collaborations (QIC)

Four (4) articles reported on the importance of quality assessment in NBS programe for sickle cell disease. It was noted that Quality improvement collaborations can provide a guiding framework for process changes aimed at improving the care of patients with sickle cell disease in busy primary care practices. Though many genetic diseases have particular health supervision guidelines, it is unknown how frequently paediatricians follow these or the best way to guarantee guideline implementation.

The American Academy of Paediatrics' Quality Improvement Innovation Networks engaged 13 practices from 11 states in a six-month QIC that provided regular educational opportunities, access to specialists in genetics, and performance evaluation [18]. For four 4 of the seven 7 targets achieving minimal data submission levels, statistically significant gains in adherence were seen. There should be in place an active structure that offers oversight of screen-positive babies identified by the Screening Program as they are handed over from screening to clinical services [19]. In Ghana, a mobile application (App) has been created, resulting in more efficient follow-up and faster time of treatment [11].

## Efficient longitudinal data capture

Eight articles pointed out the importance of data capture and management as a tool for dealing with sickle cell disease burden in various countries. The various topics discussed were: data quality, hybrid record keeping, data for policy, case specific data and longitudinal registry. Lewis Hsu et al [24] noted that availability of data about frequency, clinical course, morbidity, mortality adequate record of birth and all deliveries, are resources to convince African governments and donor agencies to commit on a comprehensive care and management plan. Results of screening tests and case-specific data are growing along with NBS programs. For accurate patient monitoring and program evaluation, the rapid acquisition of screening and other case-specific data is essential [11]. Robust Data collection and management are important to support workflows; registering babies that underwent testing, storing laboratory results, and keeping records of when families were notified of results.

## Targeted NBS for SCD

Six articles discussed the importance of targeted NBS as a strategy to minimize cost. Only screen children born by pregnant women with sickle cell trait (SCT). Another idea was to solely screen at-risk infants born to parents with SCT to adopt NBS for SCD in settings with limited resources. Health professionals thought that this strategy would be more cost-effective in reducing screening expenses. Through the prenatal clinic, sickle cell trait-carrying mothers will be identified. Newborns whose mothers have sickle cell trait will only be screened for it at delivery [22]. The ministry of health found that young children frequently have the sickle trait, supporting the significant probability of SCD that had previously been predicted. In areas with higher sickle cell trait (SCT) prevalence, which indicates a higher population illness risk, SCD screening first started [21].

### Advocacy

Seven (7) articles pointed out the need for advocacy as a means of educating all stakeholders for the expansion and scaling up NBS for sickle cell disease. To raise public awareness and educate the general public about SCD, patient groups engage in advocacy. In the USA, SCD Screening was a response to mounting political pressure by African American Advocacy groups in the 1960's [25]. Advocacy produces a community that is aware of the disease and ready to consent to child screenings for it. Additionally, advocacy involves speaking with politicians and policymakers [22]. Community engagement was noted as an important determinant of success [6].

### Funding

Four (4) articles reported on the various funding sources that could possibly benefit the NBS for SCD. From the very beginning of program development, NBS programs should prioritize funding and engagement from government partners. This can be achieved through advocacy which is at two levels; NBS sustainability depends on local MoH approval and support since it necessitates the integration of all system components within local (jurisdictional) geographic, economic, and political restrictions [28]. NBS interventions must be adapted to local conditions and incorporate support, scaling-up, and sustainability measures [26]. WHO has plan to fund major prevention of SCD in Africa, but various governments have to show satisfactory interest in having practicable and comprehensive care plans on ground [24].

### Engaging with private sectors

Collaborating with private healthcare facilities and diagnostic laboratories will help to continue the screening program implementation [22]. The article that was explicit on the sustainability of NBS programs for SCD listed engaging with private sectors, screening of children born by SCD pregnant women only, and advocacy, as the core elements of sustainability of NBS programs in resource-poor countries.

## Discussion

NBS implementation is difficult in low-income, SCD high-burden settings, such as those in most of African countries, despite being crucial for improving survival [29]. The high fertility rate and poverty level in Africa make NBS implementation difficult [11]. The scope, success, and history of NBS projects differ across Africa. Some programs are just getting started, while others have been available for a while. The integration of all system components within regional (jurisdictional) geographic, economic, and political restrictions is necessary for NBS sustainability, and as a result, local MoH endorsement is necessary. The importance of partnerships with more established programs like HIV, both for NBS implementation and research has been acknowledged [11] Actions are being taken in Ghana to provide a practical counseling training model in a community-based context due to the shortage of genetics-trained health professionals who are aware of NBS and SCD and can offer family support and counseling [30,31]. Other partnerships between established NBS programs and those in underdeveloped African nations have also been documented, including those in screening [32] and clinical/research [33,34].

The structure for a national NBS program has been laid in Nigeria but is being hindered by funding, the high cost of reagents, and skilled manpower among other factors [24]. The greatest barrier to the sustainability of NBS programs in Africa has emanated from their incomplete adoption into routine health systems [6]. From the very beginning of program development,

NBS programs should prioritize funding and engagement with government partners. This can be achieved through advocacy which is at two levels; raising public awareness and educating the general public about SCD, to foster a community that is aware of the illness and ready to consent to newborn screenings for it. In addition, advocacy involving speaking with politicians and policymakers. In the USA, SCD Screening started as a result of mounting political pressure by African American Advocacy groups in the 1960's [25]. The Ghanaian government has introduced a national digital address system in which each property is given a special identifying number. The adoption and use of digital addresses on a large scale would improve the tracking of newborns with positive screens. Such feat could only be achieved through government's participation. Efficient public health interventions such as expansion of comprehensive care services to primary and secondary health care facilities, and task shifting could go a long way to mitigate the problem of poor follow-up and enrolment encountered due to lack of funding for transportation. There is a need for the establishment of a quality Improvement Collaboration. This establishment will aim at providing a guiding framework for process changes to improve the care of patients with sickle cell disease. It will also help to check the activities of the primary healthcare centers. A study by Hettiarachchi & Amarasena [35], reported that short text messages (SMS) on mobile phones are effective in engaging with and following up with affected families.

With robust data management and genetic counseling, targeted newborn screening can go a long way to reduce costs. Proper data management is also required for effective patient follow-upand to convince African governments and donor agencies to commit on a comprehensive care and management plan and [24].

Strategies for support, scaling up, and sustainability should be included in NBS interventions that are tailored to local contexts. Dried blood spot is a blood sampling technique and is minimally invasive. Dried blood spot (DBS), has been used successfully in the isoelectric focusing method [36], and High-performance liquid chromatography [37]. For Sub- Saharan Africa (SSA), the development of national SCD NBS policy and protocol that are locally appropriate and acceptable while taking into account available resources is required. This procedure goes hand in hand with public involvement and advocacy to gain vital community support [12]. The fact that the bulk of the population in SSA lives in rural areas and lacks access to healthcare is one of the primary issues posing a big challenge to newborn screening programme [38]. Hb electrophoresis, iso-electric focusing, high-performance liquid chromatography (HPLC), mass spectrometry, and molecular methods are used in the current laboratory diagnosis of SCD. All of these are capital-intensive and necessitate highly qualified technical personnel as well as a reliable power source, both of which are in short supply in most resource-poor SSA nations. As a result, low-cost, reliable, and simple-to-use point-of-care testing (POCT) devices with high specificity and sensitivity in the discriminating of distinct Hb phenotypes are needed. The results of the evaluation of the application of HemoTypeSC device using dried blood sample (DBS) as a screening tool for SCD assessing its performance parameters against HemoTypeSC using fresh capillary blood could be a way of mitigating some of the challenges encountered over the years in scaling up and expansion of newborn screening in SSA [23]. Fresh capillary blood and screening at the point of care are all part of the standard HemoTypeSC protocol. To shorten the turnaround time, two or three people must be involved, which involves additional cost implications. When a battery of other tests is required as part of the newborn screening program, the standard POCT testing at mass screening centers could become a hurdle, but when DBS is collected for different disease conditions, the testing could be done at a more convenient time and accurate test results obtained [23].

Differential erythrocyte density differential mobility of Hb S and Hb A through filter paper [39], and a polyclonal antibody-based capture immunoassay [40] are the principles on which

some of the recently developed POCT devices for SCD are based. All of these methods have drawbacks, either because they require apparatus as an inherent element of the technique to attain maximal specificity and sensitivity, or because of their lack of accuracy [41], even though there is a claim that IEF has some advantages in Africa, such as requiring less frequent instrument maintenance and kit delivery, as well as the ability to produce IEF agarose gels locally at a lower cost [42,43]. In a study, the overall accuracy, specificity, and sensitivity of HemoTypeSC in identifying Hb phenotypes (AA, AS, AC, SS, SC, and CC) were evaluated across multiple Nigerian primary healthcare centers in a real-life, field setting, and it was discovered that the sensitivity and specificity of the POCT HemoTypeSC test for SCA were 93.4 percent and 99.9 percent, respectively, in optimal field conditions [44]. In this study, all hemoglobin C phenotype carriers were successfully identified. Furthermore, this study demonstrated an overall accuracy of 99.1% and identified significant issues in terms of staff training, particularly.

It is also necessary to use UN treaties and conventions on the rights of the child, which most countries have signed, to persuade policymakers to include the SCD NBS in their strategies to reduce childhood mortality. Article 24of the United Nations Convention on the Rights of the Child, for example, requires parties to "recognise the right of the child to the enjoyment of the greatest achievable quality of health" and to "minimise infant and child mortality [1,24,45].

In the USA, SCD Screening came into focus as a result of mounting political pressure by African American Advocacy groups in the 1960's thereafter, the state recognized SCD as a public health issue of significance. Barely ten years after that, the National Sickle Cell Anaemia Control act was passed by the congress in 1972, followed by the establishment of Education, screening testing, counseling, research and treatment programmes. The statewide NBS programe established in 1975 in New York, funded over 250 universal screening programmes, 41 sickle cell centers and clinics as well as 69 research grants and contracts, with numerous locally supported screening, education and counseling clinics. Subsequently, the success of evidence-based interventions; the National Institute of Health convened a conference that prompted unanimous support for universal NBS for SCD. Computerized data management was applied and by 1993, targeted screening based on race was replaced by universal screening. By 2006 all 50 states and many U. S territories had adopted Universal newborn screening for SCD with Hb SS, Hb SC, and Hb S/βthalassemia as core panel. Targeted NBS metamorphosed to Universal Newborn screening [25]

For Africa the narrative is different, the incorporation of all system components within regional geographic, economic, and political restrictions is necessary for NBS sustainability. African countries also battle burden of infectious diseases and malnutrition. There is low ratio of trained health workers to the population. There is low resource allocation available to health in national budgets, which primarily goes to areas identified as country priorities. There is the need for the government of SSA countries to recognize the complex relationship between SCD, infectious diseases and other non-communicable diseases [24]

What worked for America and other western countries, may not apply perfectly in African countries because of, political, economic and geographic differences and challenges. Nevertheless, in the history of NBS for SCD in the USA and Canada which spans many years, screening began with the recognition of SCD as an important public health issue, then moved to the identification of hemoglobinopathies from the same dried blood spot used to screen for other congenital disorders. This trend can be followed in African countries as described earlier where Six (6) articles advocated A hub-based collection of specimen and result delivery, for HIV, Malaria, malnutrition, other congenital diseases and SCD for cost effectiveness.

## Conclusion

Low-income, high-burden countries have so far been unable to get beyond pilot programs due to the enormous cost of developing long-term newborn screening programs, which require expensive, complicated technology not generally cheap or available, outside of large urban areas. For the sustainability of NBS programs, funding, and engagement from government partners from the very beginning of program development, should be prioritized, and screening should be tailored to the local context, More SCD patients' enrolment and follow-up should be encouraged by the expansion of comprehensive care services to primary and secondary health care facilities. With robust data management and genetic counselling, targeted newborn screening can go a long way in reducing costs. In the USA, aggressive advocacy triggered a revolution in Screening for SCD, there is much work for Advocacy groups for SCD in Africa. However,the implementation of the use of dried blood spots on HemoTypeSC could be a game changer in terms of scaling up and expanding the newborn screening program in Sub-Saharan Africa.

### Limitations of the study

Found only a few studies on the sustainability of newborn screening in Africa. This made it difficult to compare and contrast the different sustainability approaches in different parts of Africa.

Commercial or financial relationships that could be construed as a potential conflict of interest.

## Supporting information

**S1 Checklist. PRISMA-P (Preferred Reporting Items for Systematic review and Meta-Analysis Protocols) 2015 checklist: Recommended items to address in a systematic review protocol\*.**
(DOC)

**S1 Table. Summerized CASP table.**
(DOCX)

## Author Contributions

**Conceptualization:** Chinwe O. Okeke.

**Data curation:** Chinedu Okeke.

**Formal analysis:** Chinwe O. Okeke.

**Investigation:** Chinwe O. Okeke, Silas Ufelle.

**Project administration:** Obiageli E. Nnodu.

**Resources:** Chinedu Okeke.

**Software:** Chinedu Okeke.

**Supervision:** Obiageli E. Nnodu.

**Validation:** Obiageli E. Nnodu.

**Visualization:** Silas Ufelle.

**Writing – original draft:** Chinwe O. Okeke.

**Writing – review & editing:** Samuel Asala, Akinyemi O. D. Ofakunrin, Silas Ufelle.

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
