## [Decision Letter · Decision Letter 0]

30 Jan 2024

PONE-D-23-21089Sustainability of newborn screening for sickle cell disease in resource-poor countries: a systematic reviewPLOS ONE

Dear Dr. Okeke,

Thank you for submitting your manuscript to PLOS ONE. After careful consideration, we feel that it has merit but does not fully meet PLOS ONE’s publication criteria as it currently stands. Therefore, we invite you to submit a revised version of the manuscript that addresses the points raised during the review process.

We look forward to receiving your revised manuscript.

Kind regards,

Ibrahim Sebutu Bello, MBBS, MPH, MD, FMCGP

Academic Editor

PLOS ONE

Journal Requirements:

3. Please include your tables as part of your main manuscript and remove the individual files. Please note that supplementary tables (should remain/ be uploaded) as separate "supporting information" files

Reviewers' comments:

Reviewer's Responses to Questions

**Comments to the Author**

1. Is the manuscript technically sound, and do the data support the conclusions?

Reviewer #1: Yes

Reviewer #2: Partly

2. Has the statistical analysis been performed appropriately and rigorously? 

Reviewer #1: I Don't Know

Reviewer #2: No

3. Have the authors made all data underlying the findings in their manuscript fully available?

Reviewer #1: Yes

Reviewer #2: Yes

4. Is the manuscript presented in an intelligible fashion and written in standard English?

Reviewer #1: Yes

Reviewer #2: No

5. Review Comments to the Author

Reviewer #1: Manuscript Review and Evaluation

The manuscript provides a comprehensive overview of the challenges and potential strategies for sustainable implementation of Newborn Screening (NBS) programs for Sickle Cell Disease (SCD) in low-income, high-burden regions, particularly in African countries. The study amalgamates diverse sources, offering a detailed insight into the multifaceted issues hindering the efficacy of NBS programs and proposes various approaches to overcome these hurdles.

Strengths:

Comprehensive Analysis: The manuscript covers a broad spectrum of challenges, from economic limitations to inadequate resources and geographic constraints, providing a holistic view of the obstacles.

Solutions-Oriented Approach: It suggests pragmatic and contextually relevant solutions, emphasizing early government engagement, community advocacy, technological utilization, and policy alignment.

Integration of Multiple Sources: The inclusion of diverse references and studies enriches the manuscript, supporting the discourse with extensive evidence.

Areas for Consideration:

Limited Comparative Analysis: The scarcity of studies on NBS sustainability in Africa hinders the ability to present comparative sustainability approaches across different regions. Further analysis in this aspect could enhance the manuscript's depth.

In-Depth Case Studies: Incorporating detailed case studies or real-life examples of successful NBS implementation could strengthen the practicality of the proposed strategies.

Concluding Insights: The conclusion, though comprehensive, would benefit from a more specific delineation of actionable steps for immediate implementation.

Overall Assessment:

The manuscript offers a comprehensive understanding of the challenges and potential strategies for sustainable NBS implementation in low-income, high-burden regions. Despite limitations in comparative analysis and the need for more concrete conclusions, its relevance and the significance of proposed solutions make it a valuable resource in the realm of public health research.

Recommendation:

Consider further comparative analysis, inclusion of practical case studies, and a more specific and actionable conclusion to enhance the manuscript's impact and readability. With these enhancements, the manuscript will be a strong contribution to the field of public health research.

Reviewer #2: This manuscript is primarily a systematic review to assess newborn screening sustainability for sicke cell disease in countries with limited resources. Although the content is on target, I think the analysis proposed is not upto the mark, for a number of reasons.

(a). As such there is no quantitative data analysis; it's a manuscript conducting thematic content analysis and producing a deductive framework for qualitative data. However, it was strange to find that the authors could ONLY find 9 studies to be eligible. Previous research (see here: https://www.tandfonline.com/doi/full/10.1080/13645579.2015.1005453) alluded to providing some basis of sample sizes, even for conducting thematic analysis, and as such, there is no basis yet established, or mentioned in the manuscript on the validity of findings derived from analyzing only 9 studies. A well-thought-out justification is needed by the authors, in light of these (and other) published work, in order to provide some credibility of their findings.

(b) No References are provided, when CASP was introduced in Page 6.

6. PLOS authors have the option to publish the peer review history of their article (what does this mean?). If published, this will include your full peer review and any attached files.

Reviewer #1: No

Reviewer #2: No

---

## [Author Response · Author response to Decision Letter 0]

18 Mar 2024

I have attended to the points raised by the reviewers.

---

## [Decision Letter · Decision Letter 1]

24 May 2024

Sustainability of newborn screening for sickle cell disease in resource-poor countries: a systematic review

PONE-D-23-21089R1

Dear Dr. Okeke,

We’re pleased to inform you that your manuscript has been judged scientifically suitable for publication and will be formally accepted for publication once it meets all outstanding technical requirements.

Kind regards,

Ibrahim Sebutu Bello, MBBS, MPH, MD, FMCGP

Academic Editor

PLOS ONE

Additional Editor Comments (optional):

All issues raised have been resolved.

Reviewers' comments:

Reviewer's Responses to Questions

**Comments to the Author**

1. If the authors have adequately addressed your comments raised in a previous round of review and you feel that this manuscript is now acceptable for publication, you may indicate that here to bypass the “Comments to the Author” section, enter your conflict of interest statement in the “Confidential to Editor” section, and submit your "Accept" recommendation.

Reviewer #2: All comments have been addressed

Reviewer #3: All comments have been addressed

2. Is the manuscript technically sound, and do the data support the conclusions?

Reviewer #2: (No Response)

Reviewer #3: (No Response)

3. Has the statistical analysis been performed appropriately and rigorously? 

Reviewer #2: (No Response)

Reviewer #3: N/A

4. Have the authors made all data underlying the findings in their manuscript fully available?

Reviewer #2: (No Response)

Reviewer #3: (No Response)

5. Is the manuscript presented in an intelligible fashion and written in standard English?

Reviewer #2: (No Response)

Reviewer #3: Yes

6. Review Comments to the Author

Reviewer #2: (No Response)

Reviewer #3: (No Response)

7. PLOS authors have the option to publish the peer review history of their article (what does this mean?). If published, this will include your full peer review and any attached files.

Reviewer #2: No

Reviewer #3: No

---

## [Editor Report · Acceptance letter]

19 Jul 2024

PONE-D-23-21089R1 

PLOS ONE

Dear Dr. Okeke, 

I'm pleased to inform you that your manuscript has been deemed suitable for publication in PLOS ONE. Congratulations! Your manuscript is now being handed over to our production team.

Kind regards, 

on behalf of

Dr. Ibrahim Sebutu Bello 

Academic Editor

PLOS ONE